# P3b Amplitude and Latency in Tic Disorders: A Meta-Analysis

**DOI:** 10.3390/brainsci12121712

**Published:** 2022-12-14

**Authors:** Yue Yang, Hua Yang, Yao Deng, Tao Yu, Rong Luo

**Affiliations:** 1Department of Pediatrics, West China Second University Hospital, Sichuan University, Chengdu 610041, China; 2Key Laboratory of Obstetric & Gynecologic and Pediatric Diseases and Birth Defects of Ministry of Education, Sichuan University, Chengdu 610041, China

**Keywords:** tic disorders, P3b, ERP, meta-analysis

## Abstract

P3b is an event-related potential (ERP) that may be abnormal in patients with tic disorders (TD), but evidence has been inconsistent. Given the possible association between P3b and TD and the need for biomarkers for TD, the primary objective of this meta-analysis was to characterize P3b in patients with TD in comparison to healthy controls (HCs). Methods: By searching PubMed, Embase, Web of Science, SCOPUS, Medline, and Google Scholar, we identified studies that compared P3b between TD patients and HCs. The amplitude and latency of P3b were then analyzed. Subgroup analyses were conducted to investigate the influence of different experimental factors on P3b indices. Result: Overall, 19 articles involving 388 cases and 414 controls were evaluated. There were no significant abnormalities in P3b amplitude and latency in TD patients. The P3b amplitude of the TD patients was significantly decreased during the oddball task, and the P3b amplitude of the adult TD patients was also significantly decreased. Conclusion: TD patients may have an abnormal P3b compared to HCs under specified conditions.

## 1. Introduction

Tic disorder (TD) is a group of neurodevelopmental psychiatric disorders characterized by involuntary unrhythmic and stereotyped movements and/or vocalizations. TD is common in children and adolescents and can be divided into Tourette syndrome (TS), persistent (chronic) movement disorders, and transient TDs [1]. TD is often associated with various comorbidities, such as attention deficit hyperactivity disorder (ADHD) [2] and obsessive-compulsive disorder (OCD) [3]. Tics and comorbidities can lead to dysfunction and reduced quality of life by affecting emotional states (e.g., anxiety or depression), sleep quality, and causing subjective discomfort (e.g., pain or injury); they may even lead to persistent social problems (e.g., bullying or isolation) [4,5].

Using brain imaging techniques, researchers found that the circuit-based neural mechanisms underlying TS were due to a combination of hyperactivated motor pathways and reduced activation in the control portions of the cortico-striato-thalamo-cortical (CSTC) circuit. This means that communication between the basal ganglia and the motor cortex is impaired [6]. Some studies have also found hyperactivity in the supplementary motor area (SMA) in TD patients and that the SMA is largely involved in the generation of tics and sensory urges [7]. With advances in neuroelectrophysiology, an increasing number of researchers have recently applied neuroelectrophysiological techniques to study the neurodynamics of motor and cognitive processes in patients with TD. Electroencephalogram (EEG) measurement is a low-cost, non-invasive method for measuring brain activity. The simplicity and millisecond resolution of brain activity, combined with standardized analytical techniques, can be used to understand physiological function and reflect pathological changes. Event-related potential (ERP) is extracted from EEG, which noninvasively records neural activity in the brain in real time [8]. ERP is an objective factor used to study information processing and cognitive brain functions, such as attention, learning, memory, and decision-making, characterized by their positive or negative polarity, latency, and high temporal resolution [8,9].

Some biological traits of TD patients have been identified, and these traits have revealed the biological mechanisms of TD to some extent [10,11,12,13]. Importantly, P300 has been suggested to be a potential electrophysiological endophenotype in psychiatry [14]. Sutton et al. first reported that an evoked potential component peaked at approximately 300 ms and called this P300 [15]. Subsequent research revealed that P300 is related to the shift and distribution of attention and salience of stimulus [15]. Scalp topography evidence indicates that P300 was produced in the parietal, frontal, and central gyrus, as well as the anterior cingulate cortex [16]. The appearance of the P300 is thought to reflect the allocation of cognitive resources to a task [17]. It is influenced, in part, by context updating in working memory [18] as well as stimulus evaluation and categorization [19,20]. Research into the exact nature of the cognitive processes behind P300 has led to several theories [21]. According to the context-updating theory, the P300 component indexes brain activities underlying the revision of the mental representation induced by incoming stimuli [22]. According to the context-closure hypothesis, P300-eliciting stimuli are the events that closed a “context,” that is, P300 is associated with a deactivation process [23]. Therefore, it is reasonable to conclude that individual differences in the P300 values obtained from a simple stimulus discrimination task provide a reliable indication of individual differences in neural electrical processing power and speed.

P300 is typically divided into two sub-components: P3a and P3b. Of these, P3a was first observed in a tri-stimulus oddball task. In this task, a rare non-target stimulus elicits a positive deflection over the front and central electrodes of subjects. P3a is thought to reflect early attention processes, such as subjects shifting the focus of their attention to new or unexpected stimuli [22]. Another subcomponent of the P300 family is P3b. In the classic oddball task, P3b responds to the target stimulus with an increase in amplitude. P3b is thought to reflect stimulus evaluation and context updating in working memory [18]. More specifically, an increase in P3b amplitude is considered to be an indicator of the attention-driven comparison process between a new event differing from a previous event still maintained in working memory, with rarer events generating larger P3b amplitudes [22]. Furthermore, P3b latency is thought to reflect the processing speed related to stimulus classification and evaluation, suggesting its role in bridging perceptual and response processing [24].

Several previous studies have investigated P3b in patients with TD and yielded inconsistent results. Lange et al. [25] found a larger cue-locked parietal P3b amplitude during a computerized Wisconsin card sorting task in TS patients than in healthy controls (HCs), indicating enhanced proactive cognitive control in TS patients. However, other studies have found that the P3b component of patients with TS has a lower amplitude than those of healthy children in oddball tasks [26,27] tasks, reflecting impairments in attention and working memory in TS patients. Regarding latency, one study found no significant differences in the latency of P3b between normal participants and TS patients [26]. However, the P3b was delayed in a Stroop Color-Word Test [28] and figure conjunction paradigm [29], which the author interpreted as evidence of a stronger attentional effort by the TS patients to obtain behavioral results similar to control subjects in easy attentional tasks. There are several potential reasons for the different results among P300 studies on TD patients. In addition to paradigms, different experimental designs can affect the results. For example, subjects’ age [30] and modality (visual vs. auditory) [31], have been demonstrated in healthy subjects. However, the exact impact of the above-mentioned factors on TD participants requires further analysis. Moreover, the small sample size of some studies may lead to unreliable experimental results.

Given the points above, especially considering that some studies lacked sufficient statistical power due to the limited sample size, we conducted a meta-analysis aimed to assess the latency and amplitude of P3b in patients with TD. Moreover, subgroup analyses were conducted to test whether the experiment type or other relevant factors can significantly influence the results. Based on previous studies, we hypothesized that both the latency and amplitude of P3b would be smaller in TD patients than HCs.

## 2. Materials and Methods

### 2.1. Literature Search

To ensure methodological rigor, we followed the Preferred Reporting Items for Systematic Reviews and Meta-Analyses (PRISMA) [32] guidelines. Common biomedical databases, including PubMed, Embase, Web of Science, SCOPUS, Medline, and Google Scholar were searched from their inception to 10 October 2022. The following search terms were used: (“Tic Disorders” OR “Tourette Syndrome” OR TD OR TS) AND (“P300” OR “P3a” OR “P3b”) AND (“Event-Related Potentials”). These steps enabled the preliminary identification of relevant studies. The references of the included studies and related reviews were then searched to minimize the possibility of missing potential source studies. The methodological quality of the studies was assessed using the Newcastle-Ottawa Quality Assessment Scale, and studies with a score of ≥6 were eligible for the meta-analysis.

### 2.2. Inclusion/Exclusion Criteria

Only studies that met the following inclusion criteria were evaluated: (1) the mean and standard deviation of the amplitude and/or latency of the P3b waveforms were reported (if actual numbers were not reported in the studies, the results were extracted from figure reports); (2) studies were published as research papers in English; (3) cross-sectional studies compared patients with TD and HCs; (4) when multiple studies from a particular research group reported a duplicate sample, the study with the most data available was selected; and (5) participants were diagnosed with TD using the Diagnostic and Statistical Manual of Mental Disorders (III, III-TR, IV, IV-TR and 5) or International Classification of Disease 10th edition criteria. Studies were excluded if (1) they were published as abstracts or reviews or if the full text was not available and (2) if the study was an animal experiment.

### 2.3. Data Extraction

The correct extraction of valid data is a key prerequisite for synthesizing and analyzing results. Two authors (Y.Y. and H.Y.) independently extracted information from all studies that met the inclusion criteria. Disagreements were resolved through a discussion between the two authors. The following variables were collected using a standardized electronic form: first author, year, sample size, age, diagnosis, diagnostic criteria, comorbidity, modality (visual vs. auditory), reaction method, paradigm, and electrode. All but nine studies measured P3b amplitudes at the midline parietal scalp (Pz) electrode to compute effect sizes. In two of these nine studies, recordings from the parietal region were used [27,28]; one study recorded the signals from the central region [33], and one study used the average of all electrode signals [34]. Two other studies, recordings from Cz [35,36], and one study used pooled posterior region recordings [37]. In addition, two studies recorded the average signals from P7 and P8 [38], P3, and P4 [39], respectively. The electrode positions are shown in Table 1. P3b amplitude and latency were averaged from the electrode data. The GetData Graph Digitizer was used to extract the results of figure reports when event results were not reported in a table or text form.

### 2.4. Data Analysis

Changes in the latency and amplitude of the P3b evoked potentials in the TD group and HCs were compared. Multiple repeated outcomes (i.e., results from different reaction methods) were combined into one independent outcome according to the formula described in Introduction to Meta-Analysis [40] to balance each study’s weight in the overall effect size calculation.

**Table 1 brainsci-12-01712-t001:** Studies on the P3b wave included in this meta-analysis.

Author	Year	Subjects (TD)	Subjects (HC)	Mean Age (TD)	Mean Age (HC)	Diagnosis	Diagnostic Criteria	Comorbidity	Modality	Reaction Method	Paradigm	Electrode (Average)
Morand-Beaulieu S [41]	2022	47	35	11.0 ± 1.7	11.4 ± 1.6	TS	DSM-IV-TR	none	Visual	Keypress	Go/NoGo task	Pz
Kloft L [36]	2019	12	21	33.6 ± 8.8	30.9 ± 7.8	TD	DSM-IV	All patients with OCD	Visual	Keypress	Stop-signal task	Cz
Petruo V [38]	2019	35	39	12.97 ± 2.52	Matched with the TS	TS	ICD-10	3 patients with ADHD and 11 with OCD	Auditory	Keypress	Go/NoGo task	P7, P8
Sauvé G [34]	2017	12	15	33 ± 9	31 ± 9	TD	DSM-IV-TR	none	Visual	Keypress	Oddball	AF4/FC4/CP6/AF2/FC2/CP2/F6/C6/P6/F4/C4/P4/F2/C2/P2
Lange F [25]	2017	23	26	32.7 ± 11.11	32.88 ± 11.23	TS	DSM-5	unknown	Visual	Keypress	Wisconsin Card Sorting Test	Pz
Morand-Beaulieu S [27]	2016	26	27	38 ± 11.9	36 ± 13.0	TS and chronic TD	DSM-IV-TR	none	Visual	Keypress	A reinforcement-based learning and reversal task	CP1/CP2/CP5/CP6/P1/P2/P3/P4/P5/P6
Morand-Beaulieu S [27]	2016	26	27	38 ± 11.9	36 ± 13.0	TS and chronic TD	DSM-IV-TR	none	Visual	Counts	A reinforcement-based learning and reversal task	CP1/CP2/CP5/CP6/P1/P2/P3/P4/P5/P6
Eichele H [33]	2016	25	35	9.87 ± 0.24	10.04 ± 0.21	TS	DSM-IV	14 patients with ADHD	Visual	Keypress	A modified Eriksen-Flanker task	FC1/FC2/Cz/CP1/CP2
Shephard E [42]	2016	18	20	13.18 ± 2.78	13.03 ± 2.9	TS	unknown	none	Visual	Keypress	A reinforcement-based learning and reversal task	Pz
Thibault G [28]	2009	15	20	37 ± 8	40 ± 12	TS	DSM-IV-TR	none	Visual	Keypress	A stimulus–response compatibility paradigm	P3, P4, CP3, CP4
Thibault G [37]	2008	14	14	32 ± 10	37 ± 13	TS	DSM-IV-TR	none	Visual	Counts	Oddball	CP3/TP7/P3/T5/O1/CP4/TP8/P4/T6/O2
Zhu Y [26]	2006	19	20	11.169 ± 1.214	11.459 ± 1.146	TS	DSM-IV	none	Auditory	Keypress	Oddball	Pz
Johannes S [43]	2003	10	10	34.4 ± 15.3	33.7 ± 13.7	TS	DSM-IV	2 patients with ADHD, 2 with OCD and 1 with both disorders.	Visual	Keypress	Stroop-paradigm	Pz
Rothenberger A [39]	2000	11	11	12.1 ± 2.5	11.5 ± 2.1	TD	DSM-III-R	All patients with ADHD	Auditory	Keypress	An auditory selective-attention task	P3, P4
Johannes S [29]	1997	12	12	32.9	Matched with the TS	TS	DSM III -R	All patients with OCD and 3 with ADHD	Visual	Keypress	Oddball	Pz
Oades RD [44]	1996	10	12	11.7 ± 2.2	10.4 ± 1.3	TS	DSM III -R	unknown	Auditory	None	Oddball	Pz
Van Woerkom (1) [45]	1994	24	24	27 ± 3.5	28 ± 3.3	TS	DSM III -R	unknown	Auditory	Keypress	Oddball	Pz
Van Woerkom (1) [45]	1994	24	24	27 ± 3.5	28 ± 3.3	TS	DSM III -R	unknown	Auditory	None	Oddball	Pz
Van Woerkom (2) [45]	1994	29	17	12.5 ± 1.52	13.4 ± 1.35	TS	DSM III -R	unknown	Auditory	Keypress	Oddball	Pz
Van Woerkom (2) [45]	1994	29	17	12.5 ± 1.52	13.4 ± 1.35	TS	DSM III -R	unknown	Auditory	None	Oddball	Pz
Drake ME [35]	1992	20	20	8–20	10–30	TS	DSM-IIIR	10 patients with ADHD and 6 with OCD	Auditory	Keypress	Oddball	Cz/A1/A2
Van Woerkom (1) [46]	1988	20	20	27	28	TS	DSM-III	unknown	Auditory	Keypress	Oddball	Pz
Van Woerkom (2) [46]	1988	20	20	27	28	TS	DSM-III	unknown	Auditory	Counts	Oddball	Pz
van de Wetering [47]	1985	6	16	28	23	TS	DSM-III	unknown	Auditory	Counts	Oddball	Pz

ADHD: Attention-Deficit/Hyperactivity Disorder; TS: Tourette Syndrome; TD: Tic Disorder; OCD: obsessive-compulsive disorder.

To measure the effect of the mean difference between the groups and as a statistical analysis method, a random effects model was used. The I^2^ statistic was used to assess heterogeneity, with a higher I^2^ statistic indicating greater heterogeneity. Slight heterogeneity was defined as I^2^ ≤30%; moderate heterogeneity, I^2^ > 30%, ≤50%; and I^2^ > 50%, severe heterogeneity. Effect sizes were calculated for the P3b amplitude or latency by dividing the mean differences by the weighted and pooled standard deviations. Pooled effect sizes (ES: Hedges’ g) and 95% confidence intervals (CI) were estimated, and forest maps were created. The direction of the effect sizes indicated whether the P3b latency or amplitude difference between the patients and HCs was prolonged/higher (positive) or shortened/lower (negative). When the pooled effect size was equal to 0 or the 95% CI range covered 0, the difference in the result was not statistically significant. Sensitivity analyses was performed to determine the sources of heterogeneity by evaluating the effect of the successive removal of individual studies on the pooled effect size. A publication bias was assessed for using the funnel plot and Egger’s regression tests.

Subgroup analyses were examined separately for age (adult and child), paradigm (oddball and other), modality (visual and auditory), comorbidity (none, yes and unknown), and reaction method (keypress, counts, and none). Age group was defined based on the range of participants’ age and classified into two groups: all participants were <18 who were classified into an <18 group; participants ≥18 who were classified into a ≥18 group; one study that recruited subjects aged under and over 18 was excluded. Paradigm was defined as oddball and other. Oddball contained the classic oddball and the three-stimuli oddball tasks and all modified paradigms based on these two and all other different kinds of paradigms were classified into another paradigm category. If a study had two repeated measured outcomes of the subgroup classification factors, they were treated as independent samples so that they were not combined (i.e., keypress and counts). All statistical meta-analyses were performed using Stata version 16.0. All tests were two-sided, and the α test level was 0.05.

## 3. Results

### 3.1. Characteristics of Individual Studies

A total of 283 studies were identified; of these, 163 studies were excluded from the preliminary evaluation. Among the remaining 120 studies, another 77 were further excluded after a detailed assessment and 25 studies were excluded because they (1) did not report data for P3b or data could not be obtained from the figures, (2) lacked a suitable control group, and (3) included duplicate data sources. One study was added after searching the references of the above articles. Eventually, 19 studies involving 388 cases and 414 controls were included (Figure 1). Table 1 presents the characteristics of the included studies.

### 3.2. Meta-Analyses

Compared to HCs, TD participants did not show significant abnormalities, with an overall standardized mean difference of −0.12 (CI = −0.59, 0.34, *p* = 0.605) for the random effects model (Figure 2). There were significant differences between studies (I^2^ = 89.4%, *p* < 0.001). A sensitivity analysis showed that the overall standardized mean difference did not change significantly after the elimination of each study (Figure 3), indicating that the results of this meta-analysis were stable. Qualitatively (by visual inspection), the funnel plot (Figure 4) was symmetric. Quantitatively, Egger’s regression tests provided no evidence of substantial publication bias for the meta-analysis of P3b amplitude (*p* = 0.746).

In the latency analysis, TD participants did not show significant abnormalities compared with the HCs, with an overall standardized mean difference of −0.10 (CI = −0.24, 0.04, *p* = 0.149) for the random effects model (Figure 5). No differences in heterogeneity were observed between the studies (I^2^ = 0%, *p* = 0.981). The sensitivity analysis indicated that the results of this meta-analysis were stable (Figure 6). There was no significant publication bias in the studies according to the funnel plots (Figure 7) and Egger’s test (*p* = 0.395).

The synthetic result showed lower P3b amplitude than HCs in subgroups of oddball paradigms (Figure 8) and adults (Figure 9). No significant effect size of P3b was found in subgroups of child, other paradigms, modality, comorbidity, or reaction method. However, severe heterogeneity was found in all of the subgroups of P3b (Table 2).

No subgroup analyses of P3b latency showed statistical significance between the TD and HCs groups. There was little heterogeneity in all the subgroups of P3b amplitude latency.

## 4. Discussion

Only a few neurophysiological biomarkers have been identified in patients with TD. This meta-analysis found that the P3b amplitude and latency showed no significant abnormalities between TS patients and HCs. The results were stable without any publication bias. However, lower P3b amplitude in TS patients was found during the oddball paradigm and in adults. To the best of our knowledge, this is the first meta-analysis to examine P3b in patients with TD. The findings provide an important summary of the neurophysiological evidence in patients with TD.

Although P3b can be induced by a variety of different conditions, their mechanisms are similar and closely related to each other. P3b latency is related to information processing, while P3b amplitude is related to the amount of information the stimulus can deliver and the number of attentional resources devoted to the task [48,49]. According to the synthsis of studies under different experimental conditions, no difference in P3b amplitude was found between the TD group and the HC group. However, the heterogeneity of this synthesis result was very high. A variety of factors, such as paradigm, target probability, time window, response method, modality, and electrode position can affect the P3b waveform [22], making the P3b results incompatible between studies. Therefore, it is difficult to interpret whether the overall results are due to currently unknown pathophysiological mechanisms of the TD, differences in processing task requirements [50], or both.

Due to the significant heterogeneity of the combined effect size, subgroup analyses were performed to investigate root causes and test for differences between each subgroup. In the paradigm subgroup, the P3b amplitude of TD patients was significantly lower than HCs during the oddball task, which was inconsistent with the overall results, and other paradigms cannot distinguish TD from HCs groups. Moreover, the P3b amplitude was lower in adult TS patients, and this was not observed in pediatric TS patients. In general, P3 amplitude decreases when fewer attentional and cognitively related neurons are aroused. However, P3 amplitude also decreases for difficult tasks, because more mental resources are devoted to improving task performance rather than information cognitive processing [51].

Previous studies on TD report an intact P3b amplitude during a Go/No-Go paradigm [38,52], a STOP stimuli paradigm [53], an oddball paradigm [36], and a modified Eriksen-Flanker task [54]. Of these paradigms, the oddball paradigm is one of the classical ERP experimental paradigms. The oddball paradigm aims to alter the pre-programmed execution of a response by introducing a rare stimulus (with a low probability of occurrence) into a sequence of frequent stimuli. When a rare stimulus appears, the subject needs to exhibit a different type of response, which will activate the prefrontal cortex to suppress the pre-programmed response [55]. This may be a dysfunction of TD patients who already experience problems with countermanding actions when they need to suppress and reset plans [34]. In addition, the perceptual sensitivity of different paradigms to TS and normal controls differs [50], which may be the reason for the inconsistency between the results of the typical subgroup analysis and the overall results.

A previous meta-analysis showed that in healthy subjects, the amplitude of P300 increased before the age of 16 and then gradually decreased, while the timing for latency differed, with 22 years as the change age point [21]. Therefore, an age subgroup analysis was performed and revealed that adult patients with TD showed lower P3b amplitude than HCs, whereas child patients with TD showed no difference in P3b amplitude compared to HCs. This finding may suggest that reduction in the P3b amplitude may be a feature of adult TD patients. The findings of reduced P3b amplitude in TD patients showing a reduction of event-related oscillations in adult TD patients suggested that adults with TD may have a reduction in working memory updating processes. A study in adult TD patients showed decreased gray matter in the anterior cingulate gyrus and the sensorimotor areas and reductions in white matter in the right cingulate gyrus [56]. The P300 reduction has been related to impairments in the gray matter of these regions [57]. Another study reported that P300 amplitude was related to white matter volumes in the prefrontal cortex and the temporoparietal junction [58]. Thus, the decrease in P300 may reflect a decrease in white or gray matter in the prefrontal cortex and sensorimotor areas of the brain, which in turn affects tic symptoms.

The findings can be explained on the neurochemical level. Glutamate (Glu) is one of the most abundant neurotransmitters in the brain, and glutamatergic neurotransmitters play a key role in brain function [59]. Several studies have supported glutamatergic system dysfunction within CSTC pathways in adult TS patients [60,61]. A proton magnetic resonance spectroscopy (^1^H-MRS) study found that the increased glutamate neurotransmission at the frontal synapses in healthy brains was associated with greater EEG responses [62]. Given these findings, our results suggest that the reduced P3b amplitude in adult TD patients was associated with impaired glutamatergic function. Furthermore, previous drug studies have reported that GABAergic (γ-aminobutyric acid) receptors were involved in regulating P3b amplitude in healthy adults using a visual oddball task [63]. However, in children and adolescents, the concentration of GABA and glutamate was significantly higher in TS groups [64,65]. Therefore, this dysregulation of GABA and glutamate may further explain our findings and support the theory that P3b may be a trait marker of adult TD patients and merit further research to determine the biological mechanisms underlying the etiology of TD.

Furthermore, reduction in P3b amplitude has been reported in a variety of psychological conditions, such as autism spectrum disorder [66], ADHD [67], and OCD [37], suggesting that P3b amplitude reduction is not specific to TD patients and that further investigation into differences in ERP between neuropsychiatric conditions is warranted. Notably, using the oddball task, Thibault et al. [37] found that patients with TS showed higher amplitudes of P3b components than HCs, as opposed to the findings of this study. However, the patients with TS+OCD showed lower amplitudes compared to HCs. The authors inferred that the results might be due to a high comorbidity rate between TS and OCD. In other words, most ERP studies either included TS participants suffering from mild to severe OCD or provided no information about the presence of such symptoms in their sample. Therefore, a comorbidity subgroup analysis was performed, and we did not find significant differences between TD patients with no comorbidity and HCs.

P3b latency is reportedly shorter in TS children with OCD [35] or ADHD [26]. P3b is involved in complex and multilevel cognitive functions, such as working memory and the allocation of attention resources. Therefore, the researchers attributed this finding to the need for greater efforts to update working memory in TS children, and this thus occupies more attentional resources. However, in this study, we found that the P3b latency of TD patients was not significantly abnormal compared with that of HCs. This was consistent with the results of Zhu et al. [26], who found that P3b latency did not seem to be widely affected in TS children without comorbidities. The reason for this result may be that multiple influencing factors result in insignificant changes in P3b in TD children, which is not easy to observe [68].

This study has some limitations that affect the interpretation of our findings. First, some of the included studies involved subjects with other mental comorbidities, some subjects were treated with medication, the treatment cycle was different, medication types were different, and the data were insufficient. Thus, it was difficult to conduct a subgroup analysis of the influence of patient characteristics on P3b. Second, in several studies we only obtained the data for averaged electrode sites, which reduced the site-specificity of the P3b results. Third, only articles published in peer-reviewed journals were used in this meta-analysis to exclude poor-quality studies. However, unpublished articles may be excluded because they lack significant results. Thus, as with other meta-analyses, there is the possibility of publication bias, in which negative studies were not published, increasing the effect size of the pooled published studies. In addition, the search scope of this study was limited to English databases only, but eligible studies in other languages may also exist.

## 5. Conclusions

There were no significant abnormalities in P3b amplitude and latency in TD patients. The P3b amplitude of the TD patients was significantly decreased during the oddball task, and the P3b amplitude of the adult TD patients was also significantly decreased. This meta-analysis provides insights for researchers interested in the P3b effect in TD. Considering the heterogeneity of patient characteristics and study methods used in the included studies, the limitations of previous studies should be overcome in further research. For example, patients with TD should be classified according to TD subtype and course of treatment and be evaluated using a uniform paradigm. In addition, it is necessary to elucidate the neural basis and neurobiological significance of P3b in future studies.

## Figures and Tables

**Figure 1 brainsci-12-01712-f001:**
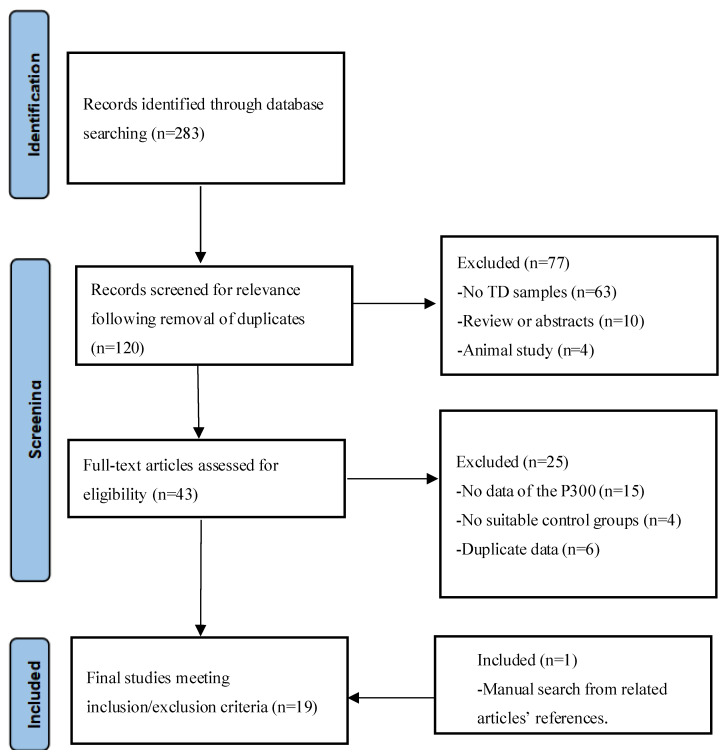
Study identification flowchart.

**Figure 2 brainsci-12-01712-f002:**
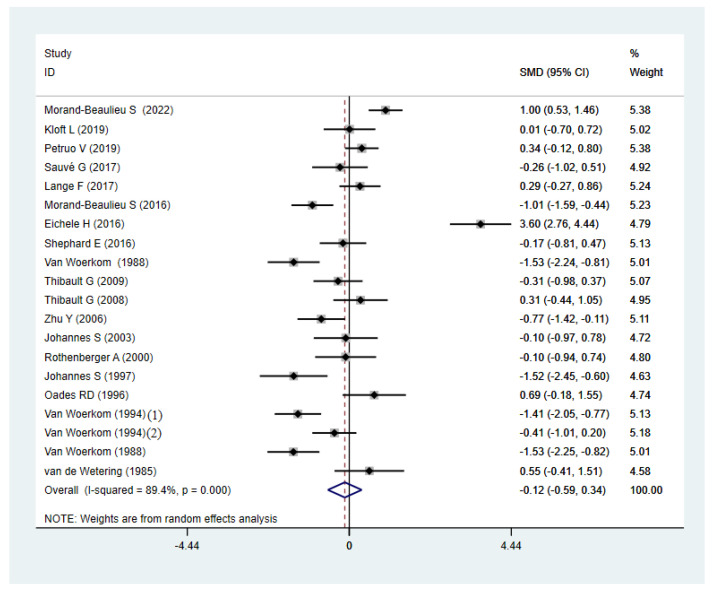
Forest map of P3b amplitude of TD patients and HCs. SMD: Standard Mean Difference; CI: confidence interval.

**Figure 3 brainsci-12-01712-f003:**
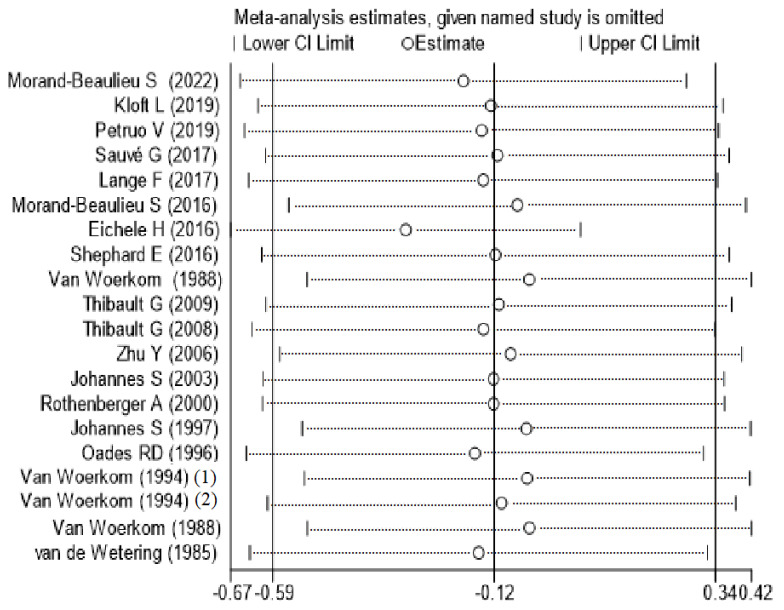
Sensitivity analysis of P3b amplitude of TD patients and HCs. CI confidence interval.

**Figure 4 brainsci-12-01712-f004:**
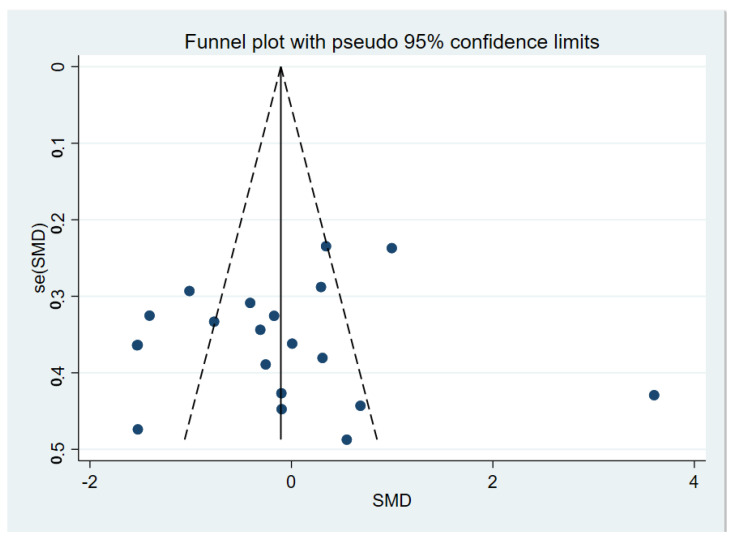
Meta-analysis on the funnel map of P3b amplitude of TD patients and HCs. SMD: Standard Mean Difference.

**Figure 5 brainsci-12-01712-f005:**
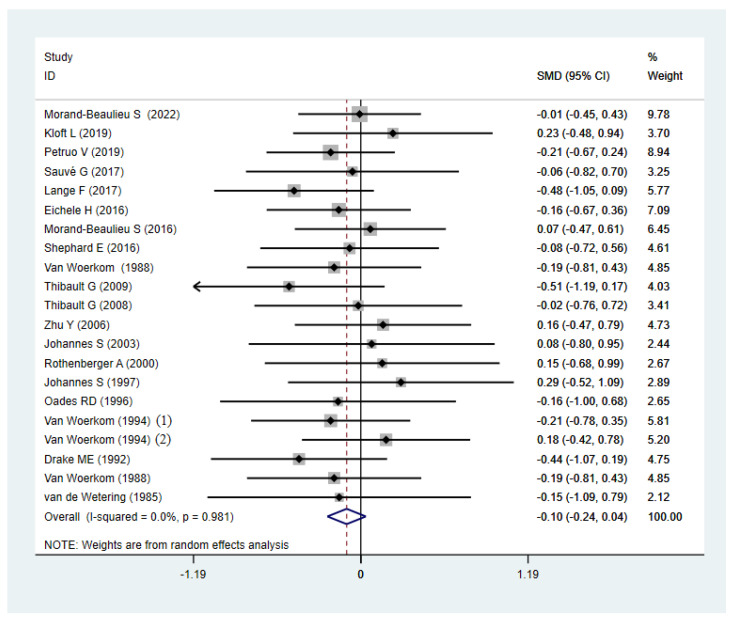
Meta-analysis on the forest map of P3b latency of TD patients and HCs. SMD: Standard Mean Difference; CI: confidence interval.

**Figure 6 brainsci-12-01712-f006:**
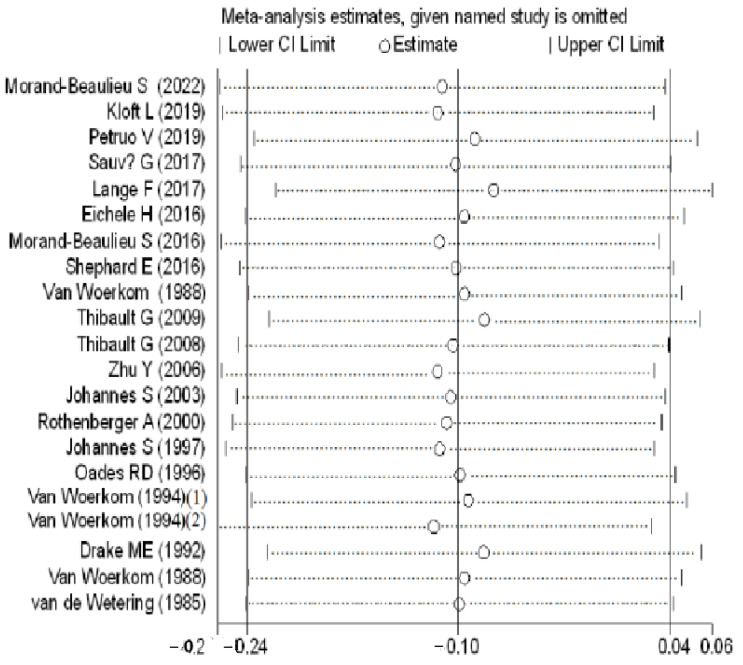
Sensitivity analysis of P3b latency of TD patients and HCs. CI: confidence interval.

**Figure 7 brainsci-12-01712-f007:**
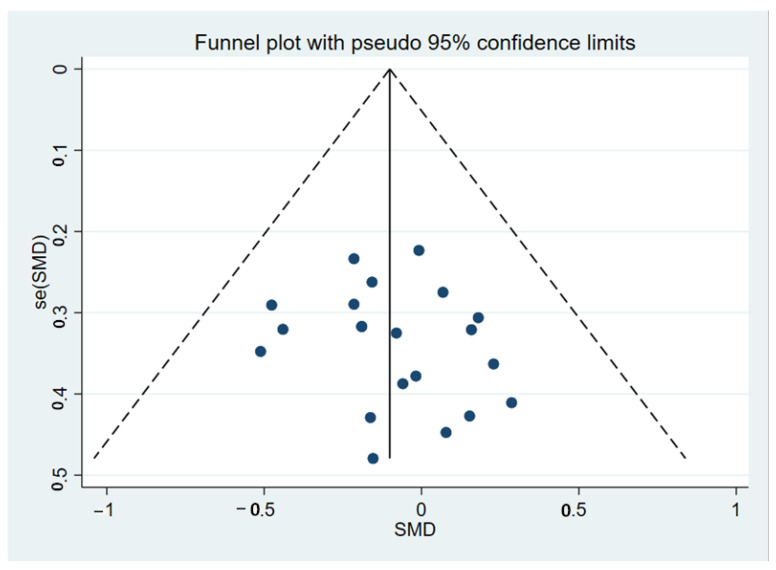
Funnel map of P3b latency of TD patients and HCs. SMD: Standard Mean Difference.

**Figure 8 brainsci-12-01712-f008:**
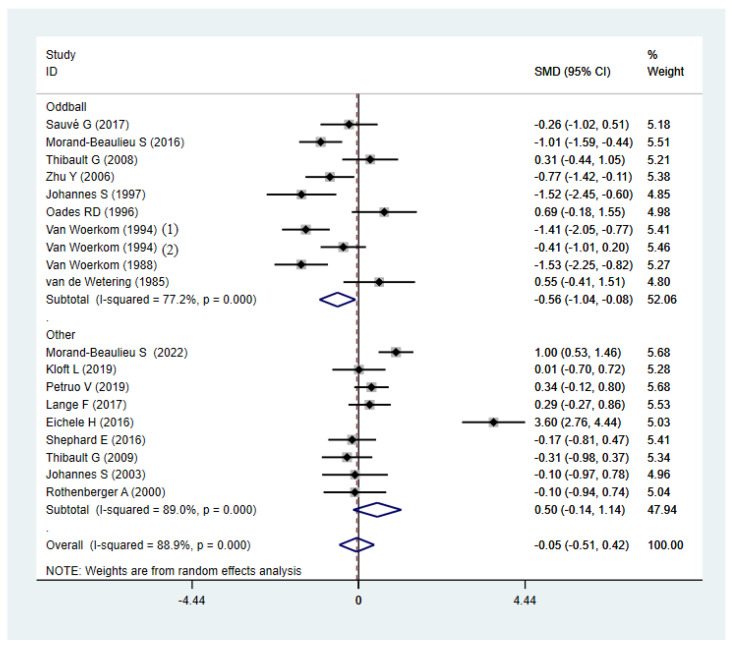
Subgroup analysis of P3b amplitude in TD patients and HCs (oddball paradigm and other paradigms). SMD: Standard Mean Difference; CI: confidence interval.

**Figure 9 brainsci-12-01712-f009:**
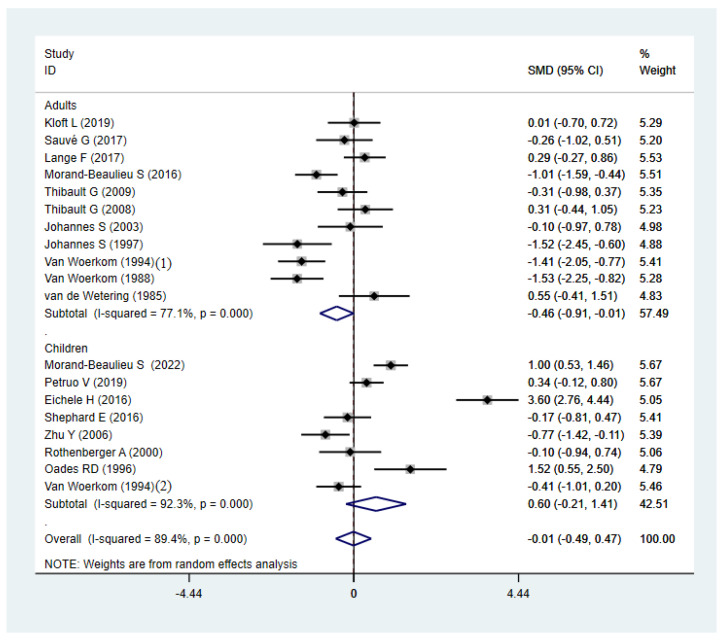
Subgroup analysis of P3b amplitude in TD patients and HCs (adults and children). SMD: Standard Mean Difference; CI: confidence interval.

**Table 2 brainsci-12-01712-t002:** P3b subgroup analysis.

	Amplitude				Latency			
	k	*p*	I2(%)	SMD (95%CI)	k	*p*	I2 (%)	SMD (95%CI)
**Age group**								
Adult	11	0.000	77.10%	−0.46 (−0.91, −0.01)	11	0.848	0.00%	−0.12 (−0.32, 0.09)
Child	8	0.000	92.30%	0.60 (−0.21, 1.41)	9	0.961	0.00%	−0.04 (−0.24, 0.16)
**Paradigm**								
Oddball	10	0.000	77.20%	−0.56 (−1.04, −0.08)	11	0.949	0.00%	−0.05 (−0.25, 0.15)
Other	9	0.000	89.00%	0.50 (−0.14, 1.14)	9	0.788	0.00%	−0.14 (−0.34, 0.06)
**Modality**								
Visual	11	0.000	90.90%	0.17 (−0.51, 0.84)	11	0.852	0.00%	−0.08 (−0.27, 0.11)
Auditory	8	0.000	85.80%	−0.26 (−0.91, 0.39)	9	0.903	0.00%	−0.12 (−0.33, 0.10)
**Comorbidity**								
None	7	0.000	83.70%	−0.16 (−0.75, 0.43)	7	0.878	0.00%	−0.04 (−0.27, 0.18)
Yes	6	0.000	93.40%	0.38 (−0.8, 1.55)	7	0.728	0.00%	−0.09 (−0.33, 0.15)
Unknown	6	0.000	88.60%	−0.20 (−1.06, 0.65)	6	0.785	0.00%	−0.18 (−0.45, 0.09)
**Reaction method**								
Keypress	16	0.000	90.50%	−0.16 (−0.69, 0.37)	17	0.906	0.00%	−0.07 (−0.22, 0.07)
Counts	4	0.000	84.60%	−0.46 (−1.39, 0.47)	4	0.935	0.00%	−0.11 (−0.44, 0.22)
None	3	0.000	93.10%	0.10 (−1.48, 1.68)	3	0.493	0.00%	−0.17 (−0.54, 0.2)

## Data Availability

Data from this study are available from the corresponding author upon request.

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
