# Peer review of "P3b Amplitude and Latency in Tic Disorders: A Meta-Analysis"

_brainsci, 2022, doi:10.3390/brainsci12121712_

Round 1
Reviewer 1 Report (Previous Reviewer 1)
I've read this new version and I think it could make a good contribution to the TS literature. Some editing for English language should be undertaken.
Two paragraphs seems odd in the discussion. The paragraphs about neurotransmitters and the paragraphs about TMS do not seem connected with the rest of the manuscript. I think it would be better to remove them.
Author Response
Thank you for this suggestion. We have removed the paragraphs about TMS. The paragraphs about neurotransmitters were intended to explain our findings at the neurochemical level. We also have revised this paragraph for better consistency with the rest article. We hope this is sufficient to ensure that this issue has now been suitably addressed.
Reviewer 2 Report (Previous Reviewer 2)
I can not see any substantial improvements to the manuscript.
Author Response
Thank you for this suggestion. Based on your comments and other reviewers, we have made major revisions to the article, including data re-inclusion, analysis, and content re-writing. I hope this manuscript is now satisfactory to you.
Reviewer 3 Report (New Reviewer)
The article well-written. I will mention a few things to improve the article
- Please report the databases in the abstract section of the paper.
- In the research method section, the exact research date is not mentioned. Please specify exactly from which year to which year this meta-analysis was searched.
Author Response
Question 1:
Please report the databases in the abstract section of the paper.
Answer to Q1:
Thank you for this feedback. We have added the databases to the abstract. Please see line 14.
Question 2:
In the research method section, the exact research date is not mentioned. Please specify exactly from which year to which year this meta-analysis was searched.
Answer to Q2:
Thank you for this suggestion. In this meta-analysis, we did not specify the exact years the search included. This is due to the small sample of appropriate studies. We wanted to include all studies that met the requirements. Therefore, the search was performed from the search engines’ inception dates to October 10, 2022.
Reviewer 4 Report (New Reviewer)
The present manuscript presents a meta-analysis of studies investigating the P3b ERP component during a standard oddball paradigm in tic disorders (TD). The authors found lower P3b amplitude in TD patients who react by keypress, and who are presented auditory stimuli. The results of the meta-analysis are put into the context of both the P3b and TD literature, and suggest that P3b may be used as a potential biomarker for TD.
This is an important topic for the field of TD research and the authors’ work is appreciated. However, there are some concerns with the validity of the results given the small number of studies included, relatively small sample sizes for subgroup analyses, and potentially important unexplored confounds that call into question the validity of results.
Major Concerns:
-
One major concern is that the generalizability of the conclusions drawn from this meta-analysis may be unsupported given the number of studies (n = 7) and subjects (n = 123 cases, n = 128 controls) included. This concern is compounded by the number of potential confounds as discussed below (e.g., stimulus modality, response method, age, comorbidities). Given the nuance involved in the results (e.g., significance reached only with auditory stimuli presentation) it is unclear whether certain results are due to the small number of studies (n=3 studies with auditory stimuli) or a real effect.
One way to increase confidence in the results would be to increase sample size by including all studies/paradigms that elicit a P300 in TD participants, rather than only studies using the “standard” oddball paradigm. For example, could you include odd-ball paradigms that fall outside this definition of standard (also, please state your definition of a “standard” odd-ball paradigm in the text)? If not, a more thorough explanation and rationale behind why non-standard oddball paradigms (n = 24) were not included in your analyses would be beneficial. What about other tasks used to compare P300 in TD vs HC, including but not limited to the stop signal task and go/no-go task?
If increasing sample size is not possible, please include a detailed explanation as to why. In this case, the description of the results throughout the abstract and discussion needs to be significantly tempered. For example, nuance must be added throughout the text to address why auditory stimuli, but not visual stimuli (and why keypress but not counts) may elicit this effect. Why is this difference meaningful or mechanistically possible? Otherwise, noisy statistics associated with insufficient sample size or an unaddressed confound seem the more likely outcome. As an example of an oversimplified/overstated statement about the results, the abstract states, “Patients with TD showed lower P3b amplitude than did HCs”. However, this is only true in the limited number of studies that met your criteria, and only in half the cases, dependent on the stimuli type and reaction method used.
-
Another concern is the influence of potential confounds, especially given the small sample. The first variable not considered in the analysis is age, which could feasibly impact findings given P300 latency and amplitude likely change with brain maturation.
https://journals.plos.org/plosone/article?id=10.1371/journal.pone.0087347. The studies used in the present analysis include both adolescents and adults, making it unclear to what extent this age effect is impacting the results. Another variable not considered is comorbid conditions. The authors mention in the Discussion a study in which participants with TD and comorbid OCD displayed results consistent with the findings of the meta-analysis. However, TD participants without comorbid OCD demonstrated higher rather than lower P3d amplitude. Given this finding, it would be helpful if the authors included a subgroup analysis using only the four studies of adults with TD and no comorbidities to increase confidence in the validity of results.
Minor Concerns:
-
On line 27, tics are referred to as “rhythmic”, whereas “unrhythmic” may be a more correct description.
-
The discussion could benefit from more coherent organization.
-
Given the mixed findings in previous studies mentioned in the introduction, it would be helpful if the authors provided further justification for their specific hypotheses.
Author Response
Question 1:
One major concern is that the generalizability of the conclusions drawn from this meta-analysis may be unsupported given the number of studies (n = 7) and subjects (n = 123 cases, n = 128 controls) included. This concern is compounded by the number of potential confounds as discussed below (e.g., stimulus modality, response method, age, comorbidities). Given the nuance involved in the results (e.g., significance reached only with auditory stimuli presentation) it is unclear whether certain results are due to the small number of studies (n=3 studies with auditory stimuli) or a real effect.
One way to increase confidence in the results would be to increase sample size by including all studies/paradigms that elicit a P300 in TD participants, rather than only studies using the “standard” oddball paradigm. For example, could you include odd-ball paradigms that fall outside this definition of standard (also, please state your definition of a “standard” odd-ball paradigm in the text)? If not, a more thorough explanation and rationale behind why non-standard oddball paradigms (n = 24) were not included in your analyses would be beneficial. What about other tasks used to compare P300 in TD vs HC, including but not limited to the stop signal task and go/no-go task?
If increasing sample size is not possible, please include a detailed explanation as to why. In this case, the description of the results throughout the abstract and discussion needs to be significantly tempered. For example, nuance must be added throughout the text to address why auditory stimuli, but not visual stimuli (and why keypress but not counts) may elicit this effect. Why is this difference meaningful or mechanistically possible? Otherwise, noisy statistics associated with insufficient sample size or an unaddressed confound seem the more likely outcome. As an example of an oversimplified/overstated statement about the results, the abstract states, “Patients with TD showed lower P3b amplitude than did HCs”. However, this is only true in the limited number of studies that met your criteria, and only in half the cases, dependent on the stimuli type and reaction method used.
Answer to Q1:
Thank you for your suggestion. We have redefined the inclusion and exclusion criteria of the articles and included other paradigms in the study. Now, 19 articles involving 388 cases and 414 controls have been included in this meta-analysis.
Question 2:
Another concern is the influence of potential confounds, especially given the small sample. The first variable not considered in the analysis is age, which could feasibly impact findings given P300 latency and amplitude likely change with brain maturation.
https://journals.plos.org/plosone/article?id=10.1371/journal.pone.0087347. The studies used in the present analysis include both adolescents and adults, making it unclear to what extent this age effect is impacting the results. Another variable not considered is comorbid conditions. The authors mention in the Discussion a study in which participants with TD and comorbid OCD displayed results consistent with the findings of the meta-analysis. However, TD participants without comorbid OCD demonstrated higher rather than lower P3d amplitude. Given this finding, it would be helpful if the authors included a subgroup analysis using only the four studies of adults with TD and no comorbidities to increase confidence in the validity of results.
Answer to Q2:
Thank you for these suggestions. Based on your first suggestion, 19 articles involving 388 cases and 414 controls have been included in this meta-analysis. To reduce the influence of potential confounds, we performed further subgroup analysis, and several factors (age, modality, comorbidity, reaction method) were included.
Question 3:
On line 27, tics are referred to as “rhythmic”, whereas “unrhythmic” may be a more correct description.
Answer to Q3:
Thank you for your suggestions. We have revised this word. Please see line 26 with tracked changes.
Question 4:
The discussion could benefit from more coherent organization.
Answer to Q4:
Thank you for your suggestion. We have revised a great deal of the discussion. We hope this revised manuscript will be satisfactory to you.
Question5:
Given the mixed findings in previous studies mentioned in the introduction, it would be helpful if the authors provided further justification for their specific hypotheses.
Answer to Q5:
Thank you for this suggestion. We have provided further justification in the introduction for the specific hypotheses. Please see lines 85-96.
Round 2
Reviewer 4 Report (New Reviewer)
I commend the authors on thoroughly addressing my concerns. I think this paper will be a nice contribution to the literature.
This manuscript is a resubmission of an earlier submission. The following is a list of the peer review reports and author responses from that submission.
Round 1
Reviewer 1 Report
In this manuscript, Yang et al. conducted a systematic review and meta-analysis of the P3 ERP component during oddball tasks in individuals with Tourette syndrome (TS). Their literature search yielded 8 studies with 123 individuals with TS and 128 controls. Meta-analyses revealed that individuals with TS had reduced P3 amplitude relative to controls, whereas P3 latency did not differ between groups.
Even though this study and its results are not groundbreaking, I think conducting such meta-analysis may be relevant. ERP studies in TS commonly have small sample, and I can only welcome any attempt to gather results from different studies in order to have more statistical power. Thus, I think this study could make an interesting contribution to the TS literature but several changes would be required
General comment
I appreciate the effort from the authors to publish their results in English, which is most likely not their native language. I think this manuscript would benefit from a good proofread by a native English speaker. However, I don’t think this really affects the quality of the manuscript and should not prevent the authors from publishing in international journals.
Title
- When I first saw the title of the paper, I feared that the authors would have lumped all TS P3 studies together, regardless of paradigms. I’m happy to see this is not the case and that a single paradigm was used. I think it may have been interesting to also include other tasks (e.g., cognitive control tasks), but it’s a good thing that these are not mixed given that the P3 in a specific task may not index the same thing as the P3 in another task. However, I think the author should mention in the title that this is a meta-analysis of the P3 amplitude and latency in TS *during oddball tasks*. As of now, it seems like this is a meta-analysis about all P3 studies in TS, which is not the case.
Introduction
- In this vein, there is no mention in the introduction that this study focuses on the oddball task. The reader learns this in the methods section. I think there needs to be a rationale for limiting the scope of this study to the oddball task.
- Apart from the fact that the P3 is the most studied ERP in individuals with TS, there is no rationale for selecting this specific ERP. In the second paragraph, the authors briefly describe the neurophysiology of TS and explain that EEG is an efficient technique to study the neurophysiology of TS. However, how the P3 is connected to that paragraph is not well explained. Why is the P3 relevant in TS and how is it connected to the neurophysiology of TS?
- In the third paragraph, the authors list the results of different studies that have assessed the P3 in individuals with TS, without really connecting the results between them (lines 56-65). Also, they list results from studies using very different paradigms, and it is not clear how those are connected to the current study. The P3 indexes (at least partially) different things in an oddball task and in a Go/NoGo task, for instance. I think that the authors should explain what the P3 indexes in different contexts.
- As a general comment for the introduction, I think it’s probably too short and many concepts would benefit from a more in-depth explanation.
Methods
- In section 2.1, the authors mention that they included both the P3a and the P3b as search terms. It is a general convention in electrophysiology that when someone talks about the P3, they are implicitly referring to the P3b. This is what I expected from the introduction. Now, if they wish to look at the P3a as well, they should discuss the subcomponents of the P3 in the introduction.
- In section 2.3, the authors mention that “electrode positions included the central parietal (Pz), frontal (Fz), and central (Cz) positions”. This is not exactly right. I’m the author of one of the papers included in the meta-analysis and we used the average of several electrodes to measure the P3, and not a single electrode. This is also the case for the papers of Thibault et al. (2008) and Sauvé et al. (2017). This sentence should thus be adapted to reflect this.
- “When the pooled effect size was equal to one or the 95% confidence interval range was one, the difference was not statistically significant.” This sentence is unclear.
- “Moderator analyses were conducted to explore the influence of the study on the P300 indices.” This sentence is unclear as well. Moderator analyses make sense here, but it is not clear which moderators were tested at this point. Also, could the authors have tested whether tic severity was a moderator?
Results
- I’m curious about the number of studies that included relevant data but had a score of less than 6 on the Newcastle-Ottawa Quality Assessment Scale.
- The abstract says that 123 individuals with TS and 128 controls are included, but according to Table 1 there are 129 individuals with TS and 144 controls.
- Two studies have assessed TS participants with two oddball tasks: one with key presses and one with a silent count of deviant stimuli (van Woerkom et al., 1988 and Morand-Beaulieu et al., 2016). The issue here is that these two studies are each given the weight of a single study, even though the same individuals performed the task. Thus, the number of participants in individual meta-analyses is inflated. I’m not an expert on how to deal with this in meta-analyses but I think that the weight of these two studies should be adapted in meta-analyses where both tasks are included (Figures 2 & 5).
Discussion
- In the discussion, the authors discuss the findings of Zhang et al. (2019) who also assessed the oddball P3 in children with tic disorders. Why wasn’t this study included in the meta-analysis?
- Lines 219-236: It is not clear what the authors aim to demonstrate here. P3 indexes different cognitive mechanisms depending on the task at hand. Thus, studies using different tasks don’t lead to inconsistent results, but allow to better understand the differences between distinct cognitive processes in TS. Also, the study by Baghai et al. (2006) is not about ERPs in children with TS. And in the next study, the family name is Eichele (Heike is her first name).
- Line 237: “Study has shown that children with TS have developmental delay [47]”. This is not true and this is not what the cited study suggest. The following paragraph also doesn’t support this idea of developmental delay in TS.
- Lines 270-272: “Subgroup analyses in the current study showed that patients with TD who reacted by keypress have lower P300 amplitude than did HCs, whereas patients with TD who reacted by counts showed no difference in P300 amplitude compared to HCs.” This sentence is unclear. It reads like participants had a choice of reacting by keypress or counts. Please rephrase so that it is clear that these are distinct protocols.
Conclusion
- This study does not provide “strong evidence”. Only 8 studies are included in the meta-analysis, with less than 150 patients. It is a good starting point, and can provide some insights for researchers interested in the P3 oddball effect in TS. However, the authors should tone down their conclusions at this point.
Author Response
Thank you for your suggestions.

Reviewer 2 Report
Yang et al.present a meta-analysis comparing P300 amplitude and latencies between patients with tic disorders and healthy controls. The authors discuss a possible link between GABA/Glutamat alterations in TD and impaired P300 amplitude. This is interesting, but should be phrased more carefully as their findings do not provide direct evidence in this regard. While the overall methodological approach seems appropriate there are some serious concerns in study design, interpretation and presentation that need to be addressed prior to publication.
Major concerns:
- The P300 has been divided in two functionally different components, the P3a and P3b. These components are most reliably differentiated by their topography with P3a peaking at fronto-central electrodes while P3b peaks at parietal electrodes. The authors decided to include data from frontal and parietal electrodes without further explanation. The authors could 1) limit their analyses to the parietal P3b 2) conduct a subgroup analyses if sufficient data is available for each group 3) show in which study P3a/P3b activity was assessed. If the authors opt for 3) this should NOT only be included in the demographic table!
- The interpretation of the funnel plots is problematic. The included studies have very similar group sizes/standard errors, which greatly diminishes the explanatory power of the funnel plots regarding publication bias. This should be clearly marked in the appropriate sections.
- Most importantly, the way some of the results are discussed seems problematic to me. The authors conclude that their findings “…suggest that TD patients have impaired attention during task execution.”. First, this interpretation is problematic as the P300 is modulated by many factors (as the authors rightfully state later on) and the relationship to attentional processes is generally linked to the P3a. Second, to conclude that TD patients from attentional impairment an ERP reduction is not sufficient, but also behavioral differences need to be apparent. The authors themselves make a statement in line with this reasoning (line 267-268).
- I do not agree with the statement that the developmental alterations in TD “refers to the delay in establishing communication between different structures, rather than the delay in the maturity of the structures themselves.”, as evidence for both factors has been presented in the last years.
- It is necessary that the authors carefully revise their references as some seem misplaced (more details in minor concerns).
- The conclusion needs to be revised. Most importantly, given the reasons stated above I do not agree that this meta-analysis provides “strong evidence”. Please specify which limitations of former studies were overcome. What do the authors mean by “treatability”?
Minor concerns:
- In the introduction there is a listing of several P300 studies (line 58 – 64). I advise the authors to rephrase and focus on the statement they want to make.
- If I am not mistaken, this should be “…equal to zero…” : “When the pooled effect size was equal to one…”
- Please identify which two authors discussed to resolve disagreement (line 102).
- I do not agree with the statement that, as of yet, specific biomarkers for TD have been found as they should have a predictive value for the individual. While this might be a rather strict definition of biomarker, I would at least tone done the statement in line 202 - 205.
- Please provide more informative captions for the figures. Abbreviations must be explained in the captions.
- Please explain abbreviations used in the text (e.g. Gln, Glu + Gln (Glx)).
- Eleven figures are way too many and should be grouped in figures with different sections (i.e. group all figures related to P300 amplitude).
- The Baghai et al. reference seems to be misplaced (ECT paper in the references).
- Heike et al. [45] should be cited as Eichele et al.
- What do the authors mean by “in ensemble averages and separate trial outcomes.” (line 230)
- Citation [48] is placed in a sentence talking about TD, but links to a P300 paper (line 239).
- I do not understand how TMS-EEG would be capable of assessing “…the dynamic activity of several neurotransmitters, including DA…”.
- Please relate the discussion in line 243 – 246 more closely to the empirical findings of this study.
- Citation [33] links to a Spanish reference?!
Author Response
Thank you for your suggestions.

Reviewer 3 Report
The authors present a meta-analysis of TD vs HC studies comparing ERP P300. The manuscript is overall well-written and clear with nice tables/figures. I do have significant concerns about the limitation of collating and interpreting the data without a better description of the co-occurring conditions present in the participants for each study. Subanalyses assessing the contribution of these diagnoses to differences in P300 values may be informative. In addition, I have the following specific recommendations/feedback:
Intro:
- Would recommend including why P300 is important and how detecting differences in P300 in TD would improve our understanding of TD. This is discussed in the discussion but would be helpful to more clearly spell out in the Intro.
Methods:
- Please describe the HC definition. Was it the same for each study? Could they have other conditions but were considered community controls? Were they age- and sex-matched?
- Please briefly describe the difference between key-press and counts in the methods section since this was a subanalysis (better orients the reader to how to think about these differently.
Figure 1:
- would like to see the specific "N" for each exclusion subcategory
Table 1:
- The mean age is presented differently for different studies. One study provides a range, one study provides SD, etc. This makes it difficult to compare. If possible, please include mean + SD (and possibly range) for each study.
- For one study, there is no reported mean age for the HC group
Figures 3 and 6
- Label X-axis
Results text:
- It would be helpful to have more information about the participants for each study, particularly in terms of the presence of co-occurring ADHD and/or OCD
Discussion:
1. One of the studies not included was cited when explaining that individuals with TD without ADHD had lower P300 amplitude compared to healthy controls. I think the possible contribution of ADHD to the results is really important and should be more fully discussed.
Author Response
Thank you for your suggestions.

Round 2
Reviewer 1 Report
I think the authors have responded correctly to my concerns and I can now recommend publication of this manuscript.
Author Response
We really appreciate the your excellent comments and suggestions that helped us to improve our manuscript.
Reviewer 2 Report
Yang et al. provided a revised version of their manuscript that did not address my most important concern. They indicated the chosen electrode in four studies, while in the remaining five studies no specific information about the electrode(s) are given. This does not represent good scientific practice. While they briefly mention P3a and P3b, interpretations linked to one or the other potential alternate throughout the discussion. Also, their subgroup analysis indicates that the amplitude change between groups was limited to the auditory oddball task, a finding that is not further discussed at all.
Author Response
Thanks for your comments.

Round 3
Reviewer 2 Report
Changes not sufficient